# What can we learn from *Harry Potter*? An Exploratory Study of Visual Representation Learning from Atypical Videos

## Abstract

Humans usually show exceptional generalisation and discovery ability in the open world, when being shown uncommonly new concepts. Whereas most existing studies in the literature focus on common typical data from closed sets, and open world novel discovery is under-explored in videos. In this paper, we are interested in asking: *what if atypical unusual videos are exposed in the learning process?* To this end, we collect a new video dataset consisting of various types of unusual atypical data (e.g. sci-fi, animation, etc.). To study how such atypical data may benefit representation learning in open-world discovery, we feed them into the model training process for representation learning. Taking out-of-distribution (OOD) detection as a task to evaluate the model's novel discovery capability, we found that such a simple learning approach consistently improves performance across a few different settings. Furthermore, we found that increasing the categorical diversity of the atypical samples further boosts OOD detection performance. These observations in our extensive experimental evaluations reveal the benefits of atypical videos for visual representation learning in the open world, together with the newly proposed dataset, encouraging further studies in this direction.

> "The most beautiful thing we can experience is the mysterious."
>
> — *Albert Einstein*

## 1 Introduction

Human cognition excels at generalising from limited information and discovering new concepts in dynamic and unpredictable environments (Lieder & Griffiths, 2020; Saxe et al., 2021). This ability to adapt to unfamiliar stimuli in an open world contrasts with the limitations faced by current machine learning models (Heigold et al., 2023), especially in the field of video understanding. Current models operate mainly in closed hypothetical environments where all possible categories are predefined during training, which limits their ability to handle the variety of unpredictable scenarios often encountered in real-world applications (Zhou et al., 2021; Kejriwal et al., 2024). The question remains whether models can be enhanced to navigate the open world with the same adaptability as human cognition.

Previous advancements in video understanding have largely focused on closed-set environments, where the model is trained and tested on well-curated (Zhu et al., 2022), typical datasets such as UCF101 (Soomro, 2012), Kinetics400, and HMDB51 (Kuehne et al., 2011). Although these models perform well within known distributions, they encounter significant difficulties when exposed to out-of-distribution (OOD) data (Acsintoae et al., 2022; Rame et al., 2022), thereby limiting their applicability to open-world environments where new and unknown categories frequently emerge (Chen et al., 2023; Ming et al., 2022). There are also ways to use generative modelling, such as GANs (Kong & Ramanan, 2021; Grcić et al., 2021) to generate virtual data or virtual features to help with OOD detection (Du et al., 2022). Existing datasets, despite being useful benchmarks, do not encourage models to generalise beyond the constraints of the training distribution (Zhang et al.,

2021). As a result, the challenge of detecting and adapting to novel instances in the open world remains an underdeveloped area in video representation learning.

The above-mentioned observations and limitations in current closed-set studies raise a crucial question: *Would that help the models' capability in open-world scenarios if introducing atypical and uncommon video data during training?* By exposing models to data that lies outside the typical distribution, we argue that it may lead to a more robust capacity for OOD detection and novel discovery (Salehi et al., 2022). Addressing this question necessitates a reconsideration of traditional video classification datasets and opens the possibility of utilising more diverse and atypical data during training.

*Atypical* data, characterised by its departure from common real-world categories, offers a unique avenue to challenge and enhance model generalisation. Unlike conventional datasets, which largely comprise trivial, everyday activities, atypical data refer to a wide range of unusual and outlier scenarios, such as those found in science fiction, animation, and anomalous real-world situations. These atypical samples present a broader spectrum of visual content, providing an opportunity for models to learn from examples that deviate from the norm (Rame et al., 2022). We anticipate that incorporating this type of data during training will allow the model to better handle open-world environments.

In order to systematically investigate the effectiveness of training with atypical data, we leverage a simple yet fundamental task – out-of-distribution (OOD) detection (Hendrycks & Gimpel, 2017). It is a critical problem in deep learning, especially in open-world settings where models are frequently exposed to data that diverges from the distribution they were trained on (Chen et al., 2023). The primary objective of OOD detection is to identify when a sample originates from an unseen or novel distribution, which is crucial for downstream tasks such as new class discovery and incremental learning (Yang et al., 2024). This capability is fundamental for models operating in open-world environments, where the ability to detect and adapt to novel inputs is critical for robust performance (Morteza & Li, 2022). An illustration is shown in Figure 1.

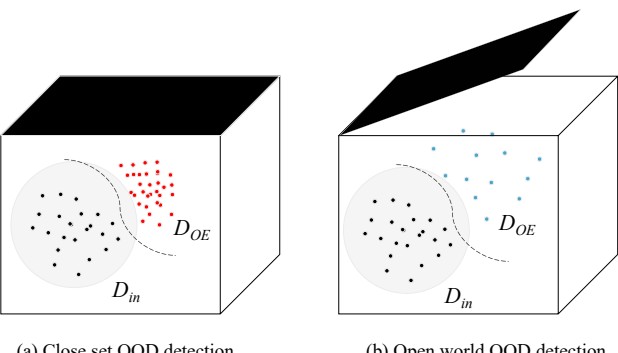

(a) Close set OOD detection  (b) Open world OOD detection

Figure 1: Illustration about the close-set OOD detection and open-world OOD detection. $D_{in}$ and $D_{OE}$ denote already known in-distribution samples and outlier exposed samples used to enhance learning capabilities, respectively. (a) denotes that the samples used for learning are still explored in a closed set despite their different distributions. (b) denotes that we open this closed set to explore a more open-world setting.

To incorporate atypical data during training, we adopt the well-established outlier exposure (OE) strategy (Hendrycks et al., 2019), which was designed to enhance models' ability to recognise OOD inputs (Papadopoulos et al., 2021; Zhu et al., 2023a; Zhang et al., 2023). The core concept behind OE is to leverage auxiliary outliers during training, enabling the model to learn to distinguish between in-distribution (ID) and OOD samples more effectively (Ming et al., 2022). However, addressing the essential distribution gap between surrogate OOD data and the unseen OOD inputs remains challenging (Zhu et al., 2023b), as it is hard to know the prior knowledge of potential OOD inputs that would be encountered at the inference stage, and intentionally collect them (Zhu et al., 2023a). Our approach seeks to mitigate this by using a diverse and atypical dataset during the training phase, aiming to better equip models to handle a wide range of potential OOD scenarios.

Extensive experiments validate the effectiveness of incorporating auxiliary outlier samples in the video domain, which significantly improves model performance. Furthermore, our analysis shows

that exposure to atypical video data (e.g. sci-fi, animation, abnormal, and unintentional) during training significantly improves the model's ability to detect OOD inputs compared to training with only traditional video datasets. Notably, we observe that the diversity of atypical samples plays a crucial role in this process. Models trained with more diverse atypical datasets show greater robustness in identifying novel and unseen distributions. These findings highlight the potential and effectiveness of the introduced atypical data in visual representation learning in the open-world setting, suggesting future investigation in this direction.

## 2 RELATED WORK

### 2.1 OPEN-WORLD LEARNING AND OOD DETECTION

Open-world learning (Kong & Ramanan, 2021; Yang et al., 2022; Vaze et al., 2021), which requires models to recognise and adapt to novel inputs, has been a key challenge. OOD detection is an essential task dedicated to handling unknown and unseen data (Yang et al., 2022). The main purpose of this task is to determine whether a sample is derived from the learned distribution $D_{in}$. A sample in $D_{in}$ is called in distribution, otherwise it is called out of distribution, denoted as $D_{out}$. The OOD distribution $D_{out}$ often simulates unknowns encountered during deployment, e.g. samples from unrelated distributions (Zhu et al., 2023a), so that the $D_{out}$ label set does not intersect with $D_{in}$ in the OOD problem setting. Out-of-distribution (OOD) detection and open set recognition (OSR) (Vaze et al., 2021; Geng et al., 2020) are closely related tasks in machine learning, both aim to deal with unknown or unseen data, but OOD is a binary classification problem that focuses more on determining whether a sample belongs to ID or OOD, whereas OSR is an additional multiclassification problem with the need to detect unknown classes (Yang et al., 2024; Salehi et al., 2022).

### 2.2 OUTLIER EXPOSURE FOR OOD DETECTION

While the test time OOD distribution $D_{out}$ remains inherently unknown (Zhu et al., 2023a), recent studies, notably by Hendrycks et al. (2019), have demonstrated the effectiveness of using $D_{aux}$ drawn from an auxiliary unlabelled dataset, to regularise the model during training. This approach leverages auxiliary outliers to encourage the model to reduce its confidence in anomalous inputs. By exposing the model to these auxiliary outliers during training, the model can better generalise to detect unknown OOD samples at test time (Hendrycks et al., 2019; Zhu et al., 2023b).

Previous studies (Ming et al., 2022; Zhang et al., 2023; Zhu et al., 2023a; Wahd, 2024) have shown that introducing auxiliary unlabelled data for OOD detection of outlier exposures in the text and image domains is very effective. However, in the same setting as the text and image domains, relatively less work has been done on OOD detection using anomaly exposure for the video domain, which may be related to the existence of a dedicated video anomaly detection (VAD) task (Sultani et al., 2018; Acsintoae et al., 2022; Nayak et al., 2021) for the video domain. However, the biggest difference between the OOD task for video action recognition and the VAD task is the difference in their purpose, where VAD is more concerned with deviations and anomalies in behaviour or patterns. In contrast, the goal of OOD for video category recognition is to expand the categories and the identification of unknown categories (Yang et al., 2024).

### 2.3 VIDEO DATASETS

Video datasets have played a crucial role in advancing computer vision research, especially in recognising human behaviour through video analysis. The success of this field has been largely due to the various video datasets released to support this research (Kuehne et al., 2011; Kay et al., 2017; Soomro, 2012; Wang et al., 2014). Most contemporary datasets are designed for tasks such as human movement classification and localisation, aiming to distinguish between various human activities (Poppe, 2010; Kong & Fu, 2022; Sun et al., 2022). Although these datasets provide benchmarks for evaluating model performance, they are limited in their representation of atypical data—rare, extreme, or fictional events that occur in real-world applications (Acsintoae et al., 2022).To address this, in this paper, we propose to explore unusual atypical data, including videos from anomaly detection, unintended actions, and fictional or animated media. We argue such atypical data is essential to open-world learning (e.g. OOD detection) in the video domain by exposing models to a broader range of variability.

## 3 ATYPICAL VIDEO DATASET

As aforementioned, we are interested in unusual atypical video data. Here we introduce, to our knowledge, the first *atypical* video dataset, consisting of various kinds of scenarios that are not common in real life. We then use this dataset for the following open-world learning study. Specifically, the dataset consists of 5,486 videos collected from existing datasets and YouTube. These clips contain both abnormal, unintentional and uncommon activities in the real world, as well as unreal video clips such as sci-fi movies and animations. Different from existing action classification and video understanding datasets, our atypical data focuses on rare/uncommon video activities, and even activities that are non-existing in the real world.

### 3.1 DATA SOURCES

The *atypical* videos dataset is composed of several subsets, each representing data that significantly deviates from typical behavioural patterns or normal visual content seen in real-world videos. These subsets include the following categories.

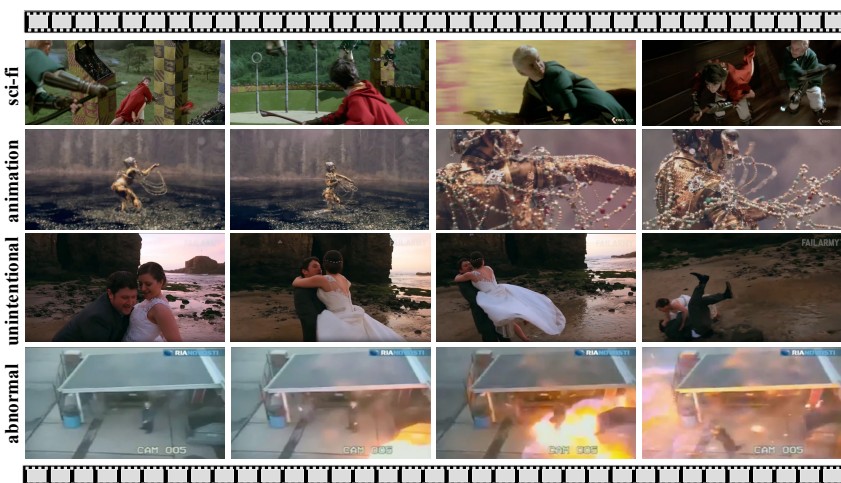

Figure 2: Examples from the proposed *atypical* video dataset.

**Sci-fi**: Sci-fi videos are collected from live-action sci-fi film trailers that are publicly available on YouTube. These clips feature futuristic or supernatural elements such as humanoid robots, space battles, or otherworldly environments. We cleaned and trimmed these videos in order to focus on targeted, action-packed, non-realistic clips that are very different from typical human behaviour. These videos differ significantly from real-world scenes in the training distribution, providing unique visual characteristics for anomaly detection.

**Animation**: In recent years, advancements in animation technology have enabled animated films to achieve a level of realism comparable to live-action footage, while simultaneously incorporating a diverse range of anthropomorphic action sequences. A notable example is Love, Death & Robots, which employ techniques such as Computer-Generated Imagery to create visually realistic yet unconventional scenarios. Additionally, trailers from widely popular animated films, such as Kung Fu Panda, have been included in our atypical dataset.

**Unintentional**: The unintentional behaviour subset is sourced from the Oops Dataset (Epstein et al., 2020), a large-scale video dataset that captures human actions involving accidental or unintentional events. We specifically used the labelled "unintentional" actions from the dataset, where the videos involve mistakes, accidents, or unexpected outcomes. By introducing this type of data, we simulate scenarios where the model may encounter unplanned or erroneous actions, enhancing its ability to handle unintended behaviours.

**Abnormal**: This subset includes videos commonly used in anomaly detection tasks. The abnormal videos are sourced from well-established video anomaly detection datasets, including Ped2 (Mahadevan et al., 2010), CUHK Avenue (Lu et al., 2013), ShanghaiTech (Luo et al., 2017), and UCF-

Crime (Sultani et al., 2018). These datasets contain surveillance footage that captures rare or unusual behaviours (e.g. accidents, criminal activities) that deviate significantly from normal actions seen in standard datasets like UCF101.

## 3.2 DATA PRE-PROCESSING

To prepare the *atypical* videos dataset for effective OOD detection, a rigorous and targeted pre-processing pipeline was implemented. Initially, all videos were manually reviewed to remove non-informative content, such as extended periods of inactivity or irrelevant scenes, ensuring focus on essential visual information. Videos were then temporally trimmed to retain action-rich segments that prominently feature *atypical* behaviours or scenarios, thus minimising redundant or extraneous frames. The selection of clips was guided by the presence of clear and distinguishable targets exhibiting behaviours significantly deviating from those seen in conventional datasets like UCF101.

Table 1: Statistical details of the proposed *atypical* video dataset.

| Subset Type | Number of Videos | Average Video Length | Key Characteristics |
| --- | --- | --- | --- |
| Sci-fi | 898 | 4.00s | Hyper-realistic, futuristic scenes |
| Animation | 859 | 4.04s | Exaggerated, non-realistic actions |
| Unintentional | 2,835 | 9.77s | Unplanned, accidental behaviour |
| Abnormal | 894 | 7.53s | Unusual, anomaly patterns |

## 3.3 DATASET STATISTICS

To ensure comprehensive coverage of anomalies in the *atypical* video dataset, we conducted a detailed analysis of the characteristics within each subset. As summarised in Table 1, we categorised the data according to its origin, content, and the diverse action scenarios it encompasses. Our dataset incorporates a wide array of scenes, targets, actions, and other elements that are typically rare in well-defined and systematically curated datasets. This diversity is further illustrated in Figure 3, highlighting the breadth of anomalous behaviours represented in the atypical samples.

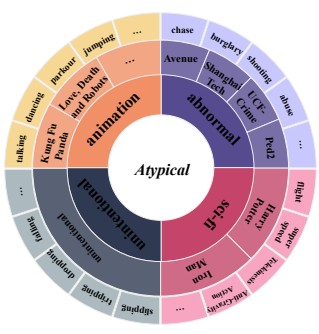

Figure 3: Illustration of the introduced *atypical* dataset composition.

## 4 HOW TO LEARN FROM ATYPICAL VIDEOS?

### 4.1 OUTLIER EXPOSURE

Out-of-distribution (OOD) detection is a critical component of open-world learning, where the goal is not only to classify known categories but also to recognise when inputs come from novel, unseen categories, enabling the system to adapt and incorporate new knowledge over time (Yang et al., 2024). It can be formulated as a binary classification problem. In the test set, the goal of the OOD detection model is to determine whether a sample $x \in X$ id from $D_{in}$ (ID) or not (OOD) (Hendrycks & Gimpel, 2017).

$$\text{OOD}(x) = \begin{cases} 1, & \text{if } P(x \mid ID) < \tau \\ 0, & \text{if } P(x \mid ID) \geq \tau \end{cases} \tag{1}$$

where $P(x \mid ID)$ denotes the probability or some confidence score that the sample $x$ belongs to the ID distribution. This is usually estimated from the softmax (Liang et al., 2018) or posterior probability (Ming et al., 2022) of the model output. $\tau$ is a pre-defined threshold to distinguish between ID and OOD data.

Since it is difficult to cover all OOD data in real-world applications, in the outlier exposure (OE) approach (Hendrycks et al., 2019; Ming et al., 2022; Zhu et al., 2023a), we introduce outlier data

$D_{out}^{OE}$ to inspire the model to find OOD signals, so as to better distinguish between in-distribution and OOD data. The goal of outlier exposure is to make the model more robust to OOD samples by learning to distinguish between normal and abnormal inputs during training (Salehi et al., 2022). Given a model $f$ and the original learning objective $L$, we can thus formalise outlier exposure as minimising the objective

$$\mathbb{E}_{(x,y)\sim\mathcal{D}_{\text{in}}}\left[\mathcal{L}(f(x),y) + \lambda\mathbb{E}_{x'\sim\mathcal{D}_{\text{out}}^{OE}}\left[\mathcal{L}_{\text{OE}}(f(x'),f(x),y)\right]\right] \tag{2}$$

Hendrycks et al. (2019) have demonstrated the effectiveness of the method in the text and image domains, and we validate its effectiveness in the video domain. From the video multi-class classification OOD task, let the input video clip be denoted as $x$, and its corresponding label as $y \in 1, 2, \ldots, k$, where $k$ is the number of action categories. The classifier is represented by the function $f : X \to \mathbb{R}^k$, such that for any input $x$, the following holds:$\mathbf{1}^\top f(x) = 1$ and $f(x) \geq 0$.

We use the Maximum Softmax Probability (MSP) (Hendrycks & Gimpel, 2017) baseline to detect OOD samples. Specifically, for a given input $x$, the model calculates the OOD score based on the maximum softmax output: OOD score $= -\max_c f_c(x)$.where $f_c(x)$ is the softmax probability for class $c$.

In the context of video classification, we perform outlier exposure by fine-tuning a pre-trained classifier $f$ so that its posterior distribution on outlier samples $D_{OE}$ becomes more uniform. The fine-tuning objective is defined as: $\mathbb{E}_{(x,y)\sim\mathcal{D}_{\text{in}}}\left[-\log f_y(x)\right] + \lambda\mathbb{E}_{x\sim\mathcal{D}_{\text{OE}}}\left[H(\mathcal{U}; f(x))\right]$, where $H$ is the cross entropy out and $\mathcal{U}$ is the uniform distribution over $k$ classes.

## 4.2 DATASETS

### 4.2.1 IN-DISTRIBUTION DATASET

**UCF101.** The UCF101 (Soomro, 2012) dataset consists of 13,320 video clips from 101 human action categories. These actions range from sports to daily activities (e.g. "biking", "swimming", "jumping"). UCF101 serves as the primary in-distribution dataset for training the model.

### 4.2.2 OUT-OF-DISTRIBUTION DATASET

**Gaussian Noise.** The Gaussian noise dataset consists of artificially generated video frames where pixel values are perturbed with noise drawn from a normal distribution $\mathcal{N}(0, \delta^2)$. This dataset is used to test the model's robustness against random noise.

**Bernoulli Noise.** This dataset is composed of binary noise, where each pixel is randomly set to 0 or 1 according to a Bernoulli distribution. It introduces a more structured yet synthetic noise pattern to challenge the model's OOD detection.

**HMDB51.** The HMDB51 (Kuehne et al., 2011) dataset contains 6,766 video clips across 51 action categories. The dataset includes a range of human activities like "punching", "climbing stairs", and "kicking". It serves as a natural OOD dataset for evaluating the model's performance on unknown human actions.

**MiT-v2**. The Moments in Time (MiT-v2) (Monfort et al., 2019) dataset includes videos covering a wide variety of events and actions not present in UCF101, such as natural phenomena and non-human actions. The dataset provides a diverse set of OOD examples, offering a broad assessment of the model's generalisation ability.

### 4.2.3 OUTLIER EXPOSURE DATASET

**Kinetics400.** The Kinetics400 (Kay et al., 2017) dataset is a large-scale video dataset widely used in the field of human action recognition. The dataset consists of approximately 240,000 video clips, each lasting approximately 10 seconds, sourced from YouTube, and is one of the most comprehensive action categorisation resources available, covering 400 different human action categories. Each video is labelled with an action category, capturing a wide range of different activities, from common actions such as "walking" and "jumping", to more complex activities such as playing a musical instrument and so on.

**The proposed *atypical*.** To further enhance OOD detection, we introduce four atypical datasets: (i) anomaly detection videos from Ped2, CUHK Avenue, and ShanghaiTech, (ii) unintentional actions from the Oops dataset, (iii) science fiction scenes sourced from movie trailers, and (iv) animated content. These diverse sources of atypical video data allow the model to learn from outliers that are visually distinct from typical action recognition datasets.

To ensure a clear distinction between intra-distributional (ID) and extra-distributional (OOD) categories, we followed the method proposed by (Hendrycks et al., 2019; Cen et al., 2023) to remove the overlap between dataset categories. Specifically, we removed 6 overlapping action categories in HMDB51 and UCF101, as well as 93 overlapping actions between Kinetics400 and UCF101 and HMDB51. In addition, 33 categories from the MiT-v2 dataset that were not present in the other three datasets were selected for testing as OOD data. Detailed information can be found in Appendix A.

This means that the categories in UCF101, HMDB51, Kinetics400, and MiT-v2 do not overlap at all in the experiment. Furthermore, the atypical dataset is significantly different from the categories in these common video datasets in terms of conceptual and visual features. By implementing category orthogonality, we effectively ensure that the OOD data are truly representative of the anomalous samples and avoid potential information leakage between the ID data and the OOD data, thus enhancing the validity and reliability of OOD detection.

### 4.3 Evaluation Metrics and Implementation Details

**Evaluation metrics.** Following the methods of (Hendrycks et al., 2019; Yang et al., 2022; Zhu et al., 2023a; Ming et al., 2022), We evaluate the OOD detection methods based on their ability to identify OOD samples, treating OOD examples as the positive class. We use three metrics: FPR95 (False Positive Rate at 95% True Positive Rate), AUROC (Area Under the ROC Curve), and AUPR (Area Under the Precision-Recall Curve). AUROC and AUPR are holistic metrics that summarise performance across multiple thresholds. FPR95 measures the false positive rate when the true positive rate is fixed at 95%, reflecting how robust the detection method is in practical scenarios. AUROC represents the probability that an OOD example receives a higher score than an in-distribution example, where a higher AUROC is better, with 50% indicating random performance. AUPR is particularly useful in imbalanced datasets with few OOD examples, as it considers the base rate of anomalies.

**Implementation details.** All experiments are based on the ResNet3D-50 (Kataoka et al., 2020) architecture as our backbone. The baseline is trained using only ID data with a cross-entropy loss for multi-class classification over 100 epochs. The initial learning rate is set to 0.1 and decays following a cosine learning rate schedule. For OOD sample testing, we use the MSP method. In the outlier exposure setting, we fine-tune the pre-trained baseline model by introducing various outlier datasets, optimising the objective function as shown in equation 2. The fine-tuning process lasts for 5 epochs. During fine-tuning, we again apply a cosine learning rate schedule with an initial learning rate of 0.001. Standard data augmentations, such as random flipping, cropping and normalisation, are applied, along with Nesterov momentum and $l_2$ weight decay with a coefficient of $5 \times 10^{-4}$.

### 4.4 Results

In this part, we evaluate the OOD detection performance using several representative outlier exposure (OE) datasets to validate the effectiveness of the proposed atypical data. Specifically, we expose various commonly used data to the baseline model to compare the impact of different OE sources. It should be noted that our Gaussian noise data and Bernoulli noise data undergo the same data enhancement and normalisation as the video data, and thus it's also a kind of OOD data worth exploring. To explore the impact of the temporal uniqueness of the video data on OOD detection, we introduce diving48 (Li et al., 2018) as $D_{OE}$ to test the performance of OOD. diving48 serves as a dataset of 48 fine-grained diving actions, which contains more than 18,000 video clips. Kinetics400 as a large-scale common action data is also used as one of the methods we compare.

As can be seen in Table 2, we present the overall results using different OE data for OOD detection. Since exposing the noisy data will allow the model to fit the pattern of out-of-distribution noisy data, the model will typically achieve better empirical performance in terms of OOD detection of noise as $D_{out}$, as reflected by the evaluation metrics. It is also for this reason that the mean metrics for AUPR

Table 2: OOD detection performance on four OOD datasets using different outlier data for outlier exposure (FPR95↓, AUROC↑, AUPR↑).

| Method | OOD Dataset | FPR95 ↓ | AUROC ↑ | AUPR ↑ |
|---|---|---|---|---|
| Baseline | Gaussian Noise | 15.95 | 87.01 | 39.26 |
| | Bernoulli Noise | 14.57 | 90.11 | 45.39 |
| | HMDB51 | 77.08 | 63.85 | 22.54 |
| | MiT-v2 | 77.73 | 64.94 | 23.76 |
| | Mean | 46.33 | 76.48 | 32.74 |
| $+OE_{Gaussian}$ | Gaussian Noise | 0.00 | 100.00 | 100.00 |
| | Bernoulli Noise | 0.00 | 100.00 | 100.00 |
| | HMDB51 | 81.11 | 63.36 | 23.03 |
| | MiT-v2 | 77.51 | 65.14 | 24.12 |
| | Mean | 39.65 | 82.13 | **61.79** |
| $+OE_{diving48}$ | Gaussian Noise | 1.06 | 99.46 | 92.74 |
| | Bernoulli Noise | 6.54 | 95.60 | 63.53 |
| | HMDB51 | 81.14 | 64.84 | 24.04 |
| | MiT-v2 | 80.87 | 65.46 | 27.24 |
| | Mean | 42.43 | 81.34 | 51.89 |
| $+OE_{K400}$ | Gaussian Noise | 7.73 | 93.53 | 54.69 |
| | Bernoulli Noise | 15.26 | 87.56 | 40.26 |
| | HMDB51 | 75.52 | 66.84 | 25.13 |
| | MiT-v2 | 67.72 | 72.53 | 30.86 |
| | Mean | 41.56 | 80.12 | 37.73 |
| $OE_{atypical}$ | Gaussian Noise | 2.99 | 97.83 | 76.14 |
| | Bernoulli Noise | 7.16 | 94.82 | 59.84 |
| | HMDB51 | 73.07 | 69.43 | 27.07 |
| | MiT-v2 | 66.62 | 74.01 | 32.59 |
| | Mean | **37.46** | **84.02** | 48.91 |

achieve the best performance of all the exposed data. However, for the real OOD datasets HMDB51 and MiT-v2, the model performance improvement is limited, suggesting that random noise makes it difficult to effectively simulate real-world complex OOD scenarios. With the introduction of Diving48 (Li et al., 2018), a fine-grained action dataset, the model's detection performance on Gaussian and Bernoulli noise was improved. However, due to the relatively homogeneous action variety of diving48, its performance improvement on the more complex realistic OOD datasets HMDB51 and MiTv2 is limited. This suggests that fine-grained data, while useful for pattern-specific learning, is not diverse enough to improve generalisation. In contrast, Kinetics400 (Kay et al., 2017) provides a wide range of action categories, and its use as OE data allows the model to perform better in all $D_{out}$ tests. This is because the data diversity of Kinetics400 helps the model learn more robust OOD detection boundaries and enhances the generalisation ability. Better performance can be obtained by exposing our atypical data for fine-tuning and then evaluating OOD detection, which validates the effectiveness of our data for probing out-of-distribution data.

## 5 WHAT CAN WE LEARN FROM ATYPICAL VIDEOS?

**Which type has the greatest impact?** To investigate this question, we conduct an ablation study by combining different categories of atypical data and evaluating their performance against various $D_{out}$ datasets. The results of combining any two categories are presented in Table 3, while further experimental results are provided in Table 4 and 5 of Appendix A. Notably, for each test dataset, we observe that nearly all category combinations, with the exception of the combination of animation and abnormal data, yield either the best or second-best OOD detection performance. Although the combination of animation and abnormal data does not always achieve the top performance, it is important to emphasise that its AUROC performance on real $D_{out}$ datasets still surpasses the baseline results. Thus, from the experimental results it is clear that for the four OOD detection

Table 3: OOD detection results across various finetuning strategies and datasets (FPR95↓ / AUROC↑ / AUPR↑).

| $D_{test}^{out}$ | Metric | $+OE_{ab\_sci}$ | $+OE_{ab\_un}$ | $+OE_{ab\_ani}$ | $+OE_{ani\_sci}$ | $+OE_{ani\_un}$ | $+OE_{sci\_un}$ |
|---|---|---|---|---|---|---|---|
| Gaussian Noise | FPR95 ↓ | 4.83 | 5.43 | 22.87 | 65.32 | 12.41 | **2.18** |
| | AUROC ↑ | 96.57 | 95.60 | 81.92 | 38.38 | 89.43 | **98.53** |
| | AUPR ↑ | 68.55 | 62.84 | 32.12 | 13.27 | 43.76 | **81.97** |
| Bernoulli Noise | FPR95 ↓ | 35.72 | 6.64 | 42.93 | 71.52 | 12.59 | **3.13** |
| | AUROC ↑ | 72.35 | 94.89 | 63.31 | 31.48 | 89.98 | **98.13** |
| | AUPR ↑ | 24.20 | 59.84 | 19.75 | 12.19 | 44.97 | **79.28** |
| HMDB51 | FPR95 ↓ | 81.86 | 78.36 | 80.72 | 82.06 | 79.06 | **76.77** |
| | AUROC ↑ | 65.50 | 65.77 | 66.63 | 66.03 | 68.09 | **69.97** |
| | AUPR ↑ | 29.38 | 23.39 | 28.40 | **30.49** | 25.93 | 28.67 |
| MiT-v2 | FPR95 ↓ | 79.98 | 68.26 | 78.49 | 79.70 | **61.40** | 64.83 |
| | AUROC ↑ | 63.89 | 73.46 | 67.23 | 65.35 | **75.30** | 74.87 |
| | AUPR ↑ | 24.36 | 31.82 | 26.91 | 25.36 | 32.94 | **33.16** |
| Mean | FPR95 ↓ | 50.59 | **39.67** | 56.26 | 74.65 | 41.36 | 36.73 |
| | AUROC ↑ | 74.58 | 82.43 | 69.77 | 50.31 | 80.70 | **85.38** |
| | AUPR ↑ | 36.62 | 44.47 | 26.80 | 20.33 | 36.90 | **55.77** |

datasets we tested, the combination of atypical categories from different data sources allows for better and more consistent OOD detection performance. Although animation data and sci-fi data contain a large amount of virtual data, they can achieve better performance when combined with abnormal and unintentional datasets, which are composed of real-world events.

**Categorical diversity of the atypical samples.** In this experiment, we incorporated various categories of atypical data, and the results are presented in Figure 4 and Figure 5. In Figure 4, each sub-figure, from left to right, represents a sequential increase in the number of atypical categories. It can be observed that the OOD detection performance generally improves as the number of atypical categories increases. A similar trend is evident in Figure 5, where we also note a progressive increase in the stability of OOD detection across different test datasets as the categorical diversity of the atypical samples expands.

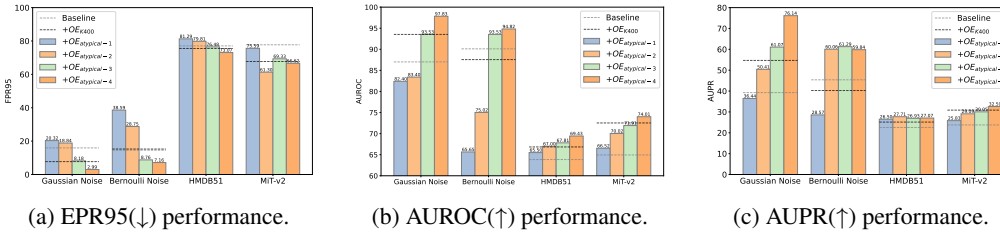

(a) EPR95(↓) performance.  (b) AUROC(↑) performance.  (c) AUPR(↑) performance.

Figure 4: Result of the effect of the number of categories of atypical data on the performance of OOD detection. atypical-n corresponds to the results for n categories in atypical outlier exposure data only, respectively.

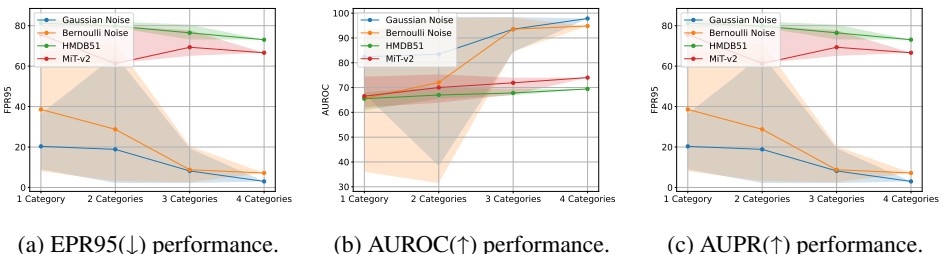

(a) EPR95(↓) performance.  (b) AUROC(↑) performance.  (c) AUPR(↑) performance.

Figure 5: Result of the effect of the number of categories of atypical data on the performance of OOD detection. atypical-n corresponds to the results for n categories in atypical outlier exposure data only, respectively.

**Closeness of $D_{out}^{test}$, $D_{out}^{OE}$, and $D_{in}^{test}$.** In this study, we utilise t-SNE to visualise the feature distributions of different datasets, as illustrated in Figure 6, to examine the relationships between $D_{in}$, $D_{out}$, and $D_{OE}$ and to explore the impact of outlier exposure data on OOD detection performance. The visualisation results indicate that UCF101, as the $D_{in}$ dataset, forms distinct feature clusters. In contrast, MiT-v2, representing $D_{out}$, displays a markedly different feature distribution from UCF101, owing to its broader range of action categories and more diverse scenarios. Additionally, the feature distributions of noisy data (Gaussian noise, Bernoulli noise) exhibit statistical properties that are more aligned with real data, likely due to similar regularisation and data augmentation processes. This similarity increases the challenge of detecting noisy data as OOD samples, highlighting the complexities involved in distinguishing these data types during OOD detection.

For $D_{OE}$, Kinetics400 and *atypical* data (unintentional, sci-fi, animation, abnormal) are used as the OE dataset, and their distributions in the feature space are more discrete compared to Kinetics400. This diverse feature distribution drives the model to learn a wider range of atypical feature patterns, which in turn enhances its ability to discriminate between OOD samples. In particular, the diversity of *atypical* data effectively improves the robustness of the model in the face of unseen scenarios or anomalous patterns by expanding the decision boundary of the model, verifying the key role of diverse anomalous exposure data in enhancing the performance of OOD detection.

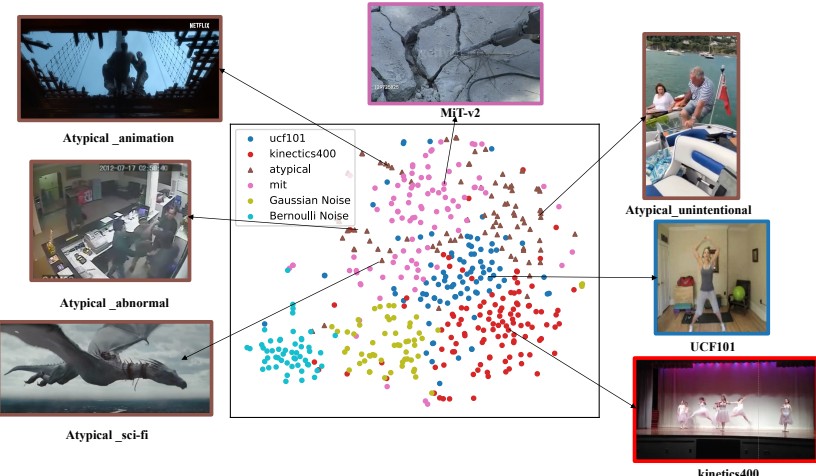

Figure 6: Feature visualisation results of $D_{in}$: UCF101; $D_{out}$: MiT-v2, Gaussian noise, Bernoulli noise; $D_{OE}$: Kinetics400, *Atypical*.

# 6 CONCLUSION

In this paper, we propose a novel dataset, termed atypical, which contains a large collection of video data that deviates from conventional, well-defined categories. This dataset was introduced to better address the challenges of open-world scenarios and to explore its impact on the critical task of OOD detection. We investigated how incorporating atypical video data enhances OOD detection in open-world settings. Our experiments suggest that training with a smaller, yet diverse set of atypical samples—such as those depicting science fiction, animation, unintentional actions, and abnormal events—substantially improves the model's robustness in identifying unseen distributions. The diversity within the atypical dataset played a crucial role in driving these improvements, underlining the importance of extending traditional datasets with more varied and unconventional content. Looking ahead, atypical data presents several promising avenues for future research. One key direction is the continued enrichment of these datasets to better capture the unpredictability of real-world environments. Furthermore, developing adaptive learning techniques that integrate new atypical samples during inference could enable models to evolve dynamically, maintaining resilience in ever-changing conditions. The integration of multimodal data, such as audio and text, with atypical video also holds the potential for enhancing models' ability to capture the complexity of open-world scenarios. Ultimately, research on atypical data opens new possibilities for advancing open-world learning and improving OOD detection.

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

# A APPENDIX

## A.1 DETAIL ABOUT DATASETS PREPROCESSING

The categories below were removed before the train. **HMDB51**: 35, Shoot bow; 29, Push up; 15, Golf; 26, Pull up; 30, Ride bike; 34, Shoot ball; 43, Swing baseball; 31, Ride horse.

**UCF101**: 2, Archery; 71, PushUps; 32, GolfSwing; 69, PullUps; 10, Biking; 7, Basketball; 6, BaseballPitch; 41, HorseRiding.

**Kinetic400**: 3, applauding; 5, arm wrestling; 18, auctioning; 19, baby waking up; 22, balloon blowing; 27, beatboxing; 31, bending back; 36, biking through snow; 40, blowing leaves; 45, bookbinding; 48, bouncing on trampoline; 49, bowling; 57, brushing teeth; 66, carrying baby; 67, cartwheeling; 68, catching or throwing baseball; 77, catching or throwing frisbee; 91, catching or throwing softball; 93, celebrating; 99, changing oil; 100, changing wheel; 101, checking tires; 102, cheerleading; 107, chopping wood; 108, clapping; 109, clay pottery making; 110, clean and jerk; 111, cleaning floor; 112, cleaning gutters; 113, cleaning pool; 114, cleaning shoes; 115, cleaning toilet; 116, cleaning windows; 117, climbing a rope; 138, climbing tree; 141, cooking chicken; 142, cooking egg; 143, cooking on campfire; 147, counting money; 148, country line dancing; 151, cracking neck; 153, crawling baby; 154, crossing river; 158, cutting pineapple; 159, cutting watermelon; 166, dancing ballet; 169, dancing gangnam style; 171, deadlifting; 174, decorating the christmas

tree; 175, digging; 176, dining; 179, disc golfing; 180, diving cliff; 182, dodgeball; 188, dribbling basketball; 220, dunking basketball; 221, dying hair; 223, eating cake; 227, eating ice cream; 230, egg hunting; 231, exercising arm; 232, exercising with an exercise ball; 237, feeding fish; 241, filling eyebrows; 246, fixing hair; 250, folding clothes; 251, folding napkins; 255, front raises; 258, gargling; 259, getting a haircut; 260, getting a tattoo; 273, giving or receiving award; 278, golf chipping; 296, grooming horse; 297, gymnastics tumbling; 305, hammer throw; 306, headbanging; 307, headbutting; 308, high jump; 309, high kick; 310, hitting baseball; 311, hockey stop; 312, holding snake; 322, hugging; 323, hula hooping; 325, ice climbing; 329, ice skating; 330, ironing; 339, javelin throw; 340, jetskiing; 345, juggling balls; 357, kissing; 367, laying bricks; 378, long jump; 395, making a sandwich; 396, writing.

And the categories of MiT-v2 below were selected.

**MiT-v2**: 2, burying; 3, covering; 4, flooding; 12, submerging; 13, breaking; 16, destroying; 17, competing; 18, giggling; 21, flicking; 34, locking; 37, flipping; 38, sewing; 39, clipping; 47, constructing; 50, screwing; 51, shrugging; 53, cracking; 54, scratching; 56, selling; 60, clinging; 87, bubbling; 88, joining; 97, kneeling; 151, peeling; 153, wetting; 159, inflating; 168, launching; 172, leaking; 205, overflowing; 221, storming; 255, combusting; 296, cramming; 297, burning.

## A.2 EXPERIMENT RESULTS

Table 4: OOD detection performance for a randomly selected atypical category.

| $D_{test}^{out}$ | Metric | $+OE_{abn}$ | $+OE_{ani}$ | $+OE_{sci}$ | $+OE_{uni}$ |
|---|---|---|---|---|---|
| Gaussian Noise | FPR95 ↓ | 13.33 | 36.44 | 22.56 | **8.97** |
| | AUROC ↑ | 88.54 | 67.34 | 81.61 | **92.13** |
| | AUPR ↑ | 41.97 | 21.66 | 31.80 | **50.31** |
| Bernoulli Noise | FPR95 ↓ | 26.61 | 66.58 | 53.13 | **8.05** |
| | AUROC ↑ | 80.78 | 36.01 | 52.37 | **93.42** |
| | AUPR ↑ | 30.84 | 12.92 | 16.24 | **54.28** |
| HMDB51 | FPR95 ↓ | 84.41 | 80.41 | **79.06** | 81.29 |
| | AUROC ↑ | 60.81 | 67.01 | **68.14** | 66.05 |
| | AUPR ↑ | 21.73 | 28.09 | **32.36** | 23.84 |
| MiT-v2 | FPR95 ↓ | 83.63 | 75.75 | 81.93 | **61.05** |
| | AUROC ↑ | 61.98 | 66.86 | 62.85 | **74.39** |
| | AUPR ↑ | 22.98 | 25.42 | 23.45 | **31.48** |
| Mean | FPR95 ↓ | 52.00 | 64.80 | 59.17 | **39.84** |
| | AUROC ↑ | 73.03 | 59.30 | 66.24 | **81.50** |
| | AUPR ↑ | 29.38 | 22.02 | 25.96 | **39.98** |

Table 5: OOD detection performance with random selection of three atypical categories.

| $D_{test}^{out}$ | Metric | $+OE_{ani\_abn\_sci}$ | $+OE_{ani\_abn\_uni}$ | $+OE_{ani\_sci\_uni}$ | $+OE_{abn\_sci\_uni}$ |
|---|---|---|---|---|---|
| Gaussian Noise | FPR95 ↓ | 19.37 | 6.81 | 4.37 | **2.16** |
| | AUROC ↑ | 84.33 | 94.60 | 96.78 | **98.42** |
| | AUPR ↑ | 35.11 | 58.68 | 69.44 | **81.05** |
| Bernoulli Noise | FPR95 ↓ | 20.46 | 7.61 | 4.37 | **2.61** |
| | AUROC ↑ | 84.58 | 94.64 | 96.95 | **98.23** |
| | AUPR ↑ | 35.50 | 59.10 | 70.56 | **80.01** |
| HMDB51 | FPR95 ↓ | 80.53 | **73.21** | 75.68 | 76.51 |
| | AUROC ↑ | 66.84 | 67.19 | **69.44** | 67.77 |
| | AUPR ↑ | **29.81** | 24.49 | 27.79 | 25.61 |
| MiT-v2 | FPR95 ↓ | 77.76 | 68.28 | **65.06** | 66.24 |
| | AUROC ↑ | 67.12 | 72.64 | 73.89 | **73.98** |
| | AUPR ↑ | 25.85 | 30.40 | 31.59 | **32.35** |
| Mean | FPR95 ↓ | 49.53 | 38.98 | 37.37 | **36.88** |
| | AUROC ↑ | 75.72 | 82.27 | 84.26 | **84.60** |
| | AUPR ↑ | 31.57 | 43.17 | 49.84 | **54.75** |

