# OpenReview forum: "What can we learn from Harry Potter? An Exploratory Study of Visual Representation Learning from Atypical Videos"
_ICLR.cc/2025/Conference — ICLR 2025 Conference Withdrawn Submission_

### Official Review · Reviewer_Gdt9 · 2024-10-24

**Soundness:** 1
**Presentation:** 2
**Contribution:** 1
**Rating:** 1
**Confidence:** 5

**Summary:**

The authors explore ways to improve out-of-distribution sample recognition (OOD) in action classification by exposing the model to diverse, out-of-distribution video samples at training time. In particular, they follow the approach of Hendrycks et al., and fine tune a pre-trained action classifier to produce uniform distribution between the classes on out-of-distribution samples. At test time, a sample a predicted as being out-of-distribution if its maximum softmax probability is bellow a certain threshold (i.e. the model is sufficiently confused, so to speak ). The contribution of this work is in comparing a few sources of out-of-distribution samples used at training and showing their effect on the models test time performance. In particular, they compare several existing datasets (Kinetics400, Oops by Epstein et al. that focuses unintentional action outcomes, a combinations of a few anomaly detection datasets as well as sci-fi and animation videos collect by the authors). The setup considers a 3D-CNN pre-trained on UCF and out-of-distribution samples come from other datasets, like MiT-v2. The results demonstrate that in this setting the combination of Oops data and either Sci-Fi or animation data does marginally better than the more conventional Kinetics400 data.

**Strengths:**

The observation that training a model to recognize out-of-distribution samples on more out-of-distribution samples improves its test time performance makes sense.

The paper is readable.

**Weaknesses:**

Although the paper is readable, the writing quality is low (grammatical mistakes, convoluted writing). Overall, the presentation quality is low (organization of the manuscript, completeness of the captions, notation, clarity, etc.).

The contribution is overclaimed in the abstract/introduction. The paper only show results on OOD (and even that in an extremely narrow setting) but claim a contribution to "visual representation learning in the open world".

The proposed "atypical" video dataset is mostly a combination of existing, public datasets. The authors collect some videos from Sci-Fi movies and animation, but seems like they will not be able to share this data (it is collected from YouTube and Hollywood movies neither of which allow re-distribution by 3rd parties). As such, there is no dataset contribution in the paper.

The dataset collection protocol is not described in sufficient detail (How the "super-categories" were selected? How the individual samples were selected for each "super-category"?) Also, the dataset is too tiny to support any representation learning claims (fewer than 6k videos).

Overview of existing video datasets doesn’t include the most recent, large-scale efforts (e.g. WebVid).

Lots of important details are missing or aren't clearly described. For example, the notation is incomplete/inconsistent: L_OE is not defined (which is the key objective in the approach), the original loss is denoted inconsistently in the equations and in the text. The notation in Table 3 and Figures 4, 5 is not defined. Gaussian noise dataset is not described in sufficient detail, which dataset is used as an original to add noise to? How exactly the amount of noise to add is determined? For some reason a new outlier dataset (diving48) is introduced inside the experiments section. It is unclear how the outlier samples are introduced during fine-tuning (e.g. is there some sample balancing between outlier and in-distribution samples?).

Outlier exposure datasets are either much larger (Kinetics400) than the in-distribution UCF-101 dataset or comparable in scale (proposed Atypical), which is not a realistic scenario. Nota that these datasets need to be labeled with action categories, because they cannot include samples from the training distribution. In practice, in a representation learning scenario, one would want to use the vast majority of the labeling effort for in-distribution data.

It is unclear why the evaluation of the effect of each datasource in Table 3 only considers pairs of data-source, and never reports the effect of each individual data source separately. On the same note, to fairly compare individual data sources, their size has to be made uniform first. Otherwise it is impossible to claim that the largest source (e.g. Oops) leads to better results because of its content, not simply because of its larger scale.

The biggest issue with this work is that the contribution seems to be minimal, if it exists at all. Is it in the observation that more diverse OOD data during training helps to better detect OOD samples at test time? This is hardly surprising/novel. Moreover, the experimental setting is too narrow to make even this unoriginal conclusion. Strictly speaking, this paper shows that using Opps + data which is very different in appearance from standard action recognition datasets (e.g. animation) is (slightly) better than using Kinetics400 when trying to learn OOD detection on UCF. And even this narrow conclusion is not clearly established because the experimental setup is somewhat flawed (see comments above). No recipy for automatically collecting/selecting useful OOD training data is provided so it is unclear how to generalize this approach to other scenarios.

**Questions:**

What is the contribution of your work?

Why did you only evaluate the effect of pairs of data sources, and not individual data sources?

What's the protocol for combining UCF with OOD data (e.g. is there sample balancing)? How was this protocol selected? Is it optimal for all studied data sources?

Do the conclusions generalize to modern model architectures (transformers)? Do they generalize to large scale datasets (e.g. using Kinetics400 as source, rathe than UCF-101)?

---

### Official Review · Reviewer_9K8M · 2024-10-27

**Soundness:** 3
**Presentation:** 3
**Contribution:** 1
**Rating:** 1
**Confidence:** 4

**Summary:**

The approach creates a video dataset combining already released video datasets and using them as known unknowns, the authors call atypical videos, for OOD classification.  The authors use the method from Hendrycks et al on using this new dataset as an outlier exposure / known unknown dataset.  The authors present ablation studies on how the different known datasets help with outlier detection.

**Strengths:**

The method tests a strategy known to work in other problems such as text and image classification on video classification to show that it works with their new dataset.

It’s nice to see different experiments on how much different outlier methods work to see how each supporting dataset separately contributes to accuracy.

The paper is easy to read and clear on what they are doing.

**Weaknesses:**

Major:
The paper is lacking in novelty and is applying known methods on known datasets.  This would fit better in an applications track at a conference rather than a general research track since there isn’t much novel about the method or the datasets.  This rise to the level of novelty required to be published at ICRL or similar conferences.

Authors need to cite Terry Boult’s work where “atypical” are called “known unknowns” and aid in detection and have been around even before this works cited here: Abhijit Bendale, Terrance E. Boult; Proceedings of the IEEE Conference on Computer Vision and Pattern Recognition (CVPR), 2016, pp. 1563-1572

Equation 2 has many undefined elements that are crucial to understanding the work.  What is LOE?  This equation is taken from the Hendrycks paper but you didn’t include any of the accompanying references to Kimin Lee, Honglak Lee, Kibok Lee, and Jinwoo Shin. Training confidence-calibrated classifiers for detecting out-of-distribution samples. International Conference on Learning Representations, 2018 who came up with the loss you are using here.  These need to be included to make this understandable.

Why did you stop at OOD rather than do OSR?  What is the benefit of not classifying the known data?  It would be interesting to explore if using this “atypical data” would hurt the known class classification to explore the tradeoffs with this kind of data.

The noise to create OOD is very close to many adversarial work to show robustness or to attack networks.  For example: Jiang, Linxi, Xingjun Ma, Shaoxiang Chen, James Bailey, and Yu-Gang Jiang. "Black-box adversarial attacks on video recognition models." In Proceedings of the 27th ACM International Conference on Multimedia, pp. 864-872. 2019.  This is related to this approach since you are using this type of noise to determine OOD.


Minor:
Line 126ish:  OSR has an OOD problem within it.  OSR is a two step process where the first step is to do OOD and then, if from a known class, classify it.  OOD could be considered an anomaly detection task as well though your definition above (Line 143) says that you are more focused on class labels.

Figure 4, please add horizontal lines.

Line 269, you are saying it is difficult but that means it is possible.  Are you actually stating this is possible for real-world applications?

**Questions:**

Line 147:  Does the frequency of the OOD class within the testing dataset make a difference here?  Typically, OOD for new classes means that the class has multiple examples within the test dataset while a kind of anomaly only has one or very few.

The atypical data here seems to be similar to the known unknowns from Terry Boult’s work (Abhijit Bendale, Terrance E. Boult; Proceedings of the IEEE Conference on Computer Vision and Pattern Recognition (CVPR), 2016, pp. 1563-1572).  How are you distinguishing from previous works like this and why are you renaming it to atypical?  Even in Hydrics works, they call it outlier exposure.  Why are you renaming it here?

How are you ensuring that the activities within the unseen data are not within the other parts of the dataset?  While you look at categories in the appendix (glad to see it), how are you avoiding very similar or the same action labeled differently or how some activities aren’t labeled within the atypical datasets?

Since you are training on more data, isn’t this an unfair comparison with the other methods?

---

### Official Review · Reviewer_37Qz · 2024-11-01

**Soundness:** 2
**Presentation:** 3
**Contribution:** 1
**Rating:** 3
**Confidence:** 5

**Summary:**

The authors propose a novel dataset containing atypical videos across 4 categories - sci-fi, animation, unintentional actions and anomalies, to fine-tune ResNet3D-50’s out of distribution detection capability. They found that introducing more categories of atypical videos further boost performance.

**Strengths:**

The first paper to introduce a dataset containing atypical videos in sci-fi and animation category.

**Weaknesses:**

1. Very limited experiments - fine-tuning only vanilla ResNet, with one in-distribution dataset and showing improvement on that is not enough at all. There are a lot of strong models in existing literature that do OOD detection with high robustness to outliers. To show effectiveness of the proposed atypical dataset, need a much more extensive experiments on stronger models and more in-distribution datasets.

    2. Missing quantitative evaluations - Randomly combining some of the 2,3 categories of atypical dataset does not give any meaningful result. To get a more meaningful performance, need to show all combinations of categories.  Moreover, the mean result across datasets is not a meaningful quantitative performance because of the difference in data distribution, performance across different such datasets cannot be averaged.

    3. The generation of the dataset is not well motivated enough. Sci-fi and animation data is non-existent in real-world scenario, so having these as OOD samples and claiming it will generalize open-world OOD detection better is too far-fetched and not supported by quantitative evaluation. Fine-tuning the model on only these categories has worse performance than baseline (Figure 4, Table 4), which again proves that introduction of these samples are not helping the model in any way.
    4. The dataset statistics is incomprehensive - important explanation about how videos were selected for unintentional and abnormal category from existing datasets, how frames were sampled, why the number and video length of unintentional category is much higher than others etc is missing. These important details about the skew in data distribution might drive a better analysis of performance for this category.
    5. The effect of fine-tuning with Gaussian noise, diving48 and K400 is not well explained. No extensive analysis provided on those datasets about how they are not enough and why atypical is a more effective OOD dataset than these for outlier exposure? Moreover, fine-tuning with Diving48 already gives much better performance than fine-tuning with atypical dataset. This invalidates the effectiveness of the proposed atypical dataset.
    6. Formatting and readability issues - what most of the symbols denote is not mentioned in table captions. Redundant figures (figure 4 and 5) that provide no new information. Extremely small font on figures and placement issues hamper readability.  Moreover, baseline performance not being present in Table 3, 4, 5 causes severe readability issues.

**Questions:**

1. How was the gaussian noise dataset generated? What was the original pixel values that were perturbed with gaussian noise? Is it gaussian noise applied on any of the existing dataset?
    2. How are the atypical-n categories (n=2,3) selected to finetune? Is there any motivation behind selecting certain combinations and not others?

---

### Official Review · Reviewer_bLey · 2024-11-02

**Soundness:** 2
**Presentation:** 3
**Contribution:** 2
**Rating:** 3
**Confidence:** 4

**Summary:**

This paper investigates the impact of atypical video data on representation learning for open-world discovery. A new dataset featuring diverse unusual video types is introduced to enhance model training. The study demonstrates that incorporating atypical data improves out-of-distribution detection performance, especially when the categorical diversity of samples is increased.

**Strengths:**

+ The paper is well-written and very easy to understand.
+ The experimental results of the paper are very good compared to the baseline.

**Weaknesses:**

- The experimental results are insufficient.
- There is a lack of insight regarding the core atypical data.

**Questions:**

- it appears that atypical video data is useful for OOD, and the attempted OE-baed methods. However, it seems that the data and methods presented in the work are independent of videos and could be adequately demonstrated in NLP, audio, or image domains as well. Why is the focus solely on video?
- The results show that there is no convergence. From the results in Fig. 4 and Fig. 5, it is evident that increasing the number of atypical categories can improve performance; why not continue to add more categories?
- The new data quality is only 5486. If the dataset increases by one order of magnitude, what would the result be?
- Regarding the atypical data distribution, quantity, categories, or other attributes, how should we define their quality? This work does not provide clear experimental conclusions. Therefore, this is an unfinished task, and I am unsure whether my understanding is correct.

---

### Note · Authors · 2024-11-14

**Comment:**

We thank all the reviewers for their constructive comments and suggestions. We will carefully consider them and improve our work accordingly.

**Withdrawal Confirmation:**

I have read and agree with the venue's withdrawal policy on behalf of myself and my co-authors.